# New Concepts of Regeneration and Renewal of Adrenal Chromaffin Cells

**DOI:** 10.3390/ijms26199369

**Published:** 2025-09-25

**Authors:** Nataliya V. Yaglova, Sergey S. Obernikhin, Svetlana V. Nazimova, Valentin V. Yaglov, Ekaterina P. Timokhina, Elina S. Tsomartova, Marina Y. Ivanova, Elizaveta V. Chereshneva, Tatiana A. Lomanovskaya, Dibakhan A. Tsomartova

**Affiliations:** 1Laboratory of Endocrine System Development, A.P. Avtsyn Research Institute of Human Morphology of Federal State Budgetary Scientific Institution “Petrovsky National Research Centre of Surgery”, 119991 Moscow, Russia; ober@mail.ru (S.S.O.);; 2Department of Human Anatomy and Histology, Federal State Funded Educational Institution of Higher Education I.M. Sechenov First Moscow State Medical University, 119435 Moscow, Russia

**Keywords:** chromaffin cells, adrenals, regeneration, self-maintenance, proliferation, transcription regulation, embryonic development

## Abstract

Chromaffin cells are neuroendocrine cells found in the adrenal medulla and paraganglia. They represent enigmatic cell population with origins and properties that have undergone a change in scientific interpretations over the last few decades. Earlier concepts consider that chromaffin cells derive from neuronal progenitors, and their cell fate is similar to neurons that lack the ability to proliferate and maintain renewal of cell population in postnatal life. Growing evidence of postnatal proliferation and response to proliferative stimuli were inconsistent with traditional views and required their reassessment and further research on chromaffin cell regeneration sources. The present review summarizes data on embryonic origin and development and transcriptional control of the adrenal chromaffin cells as well as available information about their postnatal proliferation. The authors also represent their findings in cellular and molecular events associated with the physiological transition from organ growth to self-maintenance of cell populations in intact rats and in experimental dismorphogenesis of the adrenals. The authors familiarize readers with available information about the early development and molecular changes in chromaffin cells in postnatal period and propose their new theories concerning mechanisms of adrenomedullary chromaffin cell regeneration. Further research on induction and management of these mechanisms will allow us to maintain cultured chromaffin cells in vitro, which will obviously make a significant contribution to practical regenerative medicine.

## 1. Introduction

Four main morphogenetic processes underlie the development and growth of organs: cell division, differentiation, apoptosis, and migration. The combination of these processes, both in quantitative and temporal aspects, forms a unique development program for each organ. After the completion of growth, the organs proceed to physiological regeneration, that is, the self-maintenance of cell populations.

In human and animal organisms, there are various cell populations that differ in embryonic origin, structure, function, and their ability to regenerate. Epithelial and connective tissue cells are regarded as cells with high regenerative potential, whereas nervous tissue, especially neurons, is known to have the lowest capacity for regeneration in physiological and pathological conditions. In recent years, natural and induced reparation and self-maintenance of chromaffin cells have gained great scientific and practical interest. The reason was the unique function of chromaffin cells and, thereafter, their possible application in therapy and the expansion of novel therapeutic strategies.

Chromaffin cells represent a neuroendocrine cell population that forms the parenchyma of the adrenal medulla and paraganglia. They are the main source of catecholamines, and therefore, they determine the functioning of internal organs, affect metabolic parameters, and play a key role in the implementation of the protective reaction. That is, they have a functional similarity to adrenergic neurons. This similarity allowed us to rely on the use of chromaffin cells as sources of catecholamines, dopamine, and opioid peptides in the protocols of cell therapy for neurodegenerative diseases, such as Parkinson’s disease, brain injuries, strokes, and alleviation of chronic pain [1,2,3,4,5]. A large number of transplantations of chromaffin cells to laboratory animals and patients have been performed since the 1980s, but they mostly yielded disappointing results, as the positive therapeutic effect was not prolonged and lasted 1–2 years because of the low viability of autografted cells [6,7].

Chromaffin cells are considered to be low-proliferating. There was even a view that chromaffin cells in adults are a postmitotic population that fails to renew. Later investigations have refuted this concept. In this regard, the ability of chromaffin cells to regenerate and, most importantly, to self-maintain and renew throughout life still attracts the interest of investigators. The scientific literature contains information about the ability of chromaffin cells to proliferate in postnatal development [8]. However, it is not clear how effective and unique this mechanism is for maintaining this cell population. The present review summarizes the available data on adrenal chromaffin cell origin and embryonic development, postnatal proliferation and self-maintenance, and novel concepts of chromaffin cell reparation.

## 2. The Origin of the Chromaffin Cells

Scientific views on the origin of chromaffin cells in the twentieth century have changed several times. There was a view that chromaffin cells and sympathetic neuroblasts have a common origin from the sympathoadrenal cell [9]. But, as early as the 1960s and 1970s, the Soviet histologist Nina Smitten put forward an alternative assumption about the origin of chromaffin cells from prospongioblasts, i.e., glial progenitors [10]. However, it did not receive further development, since the hypothesis about the common origin of chromaffin cells and neurons had a greater number of followers due to the discovery of the facts that, at first glance, confirmed this point of view. These facts include the ability of chromaffin cells to acquire a neuron-like phenotype when stimulated by nerve growth factor (NGF) [11], as well as the identity of the surface antigens of chromaffin cells and sympathetic neuroblasts found on the 11.5th day of embryonic development in mice [12,13].

Currently, most researchers are inclined to believe that chromaffin cells arise from neuron-associated multipotent cells, also known as Schwann cell precursors [14,15,16]. These cells migrate along the preganglionic sympathetic nerves innervating the sympathoadrenal rudiment, and then from 11.5 to 14.5 days of human embryonic development differentiate into chromaffin cells [16,17]. In mice and rats, Schwann cell precursors migrate to the adrenal gland, penetrating the cortex [18,19]. In the cortex, chromaffin cells integrate into spherical clusters, which then form the medulla. The formation of chromaffinoblasts in the adrenal glands of rats occurs from the 18th to 19th day of antenatal development until the 10th day of the postnatal period [10].

Acquisition of a specific secretory phenotype is the main event in Schwann cell precursor differentiation into chromaffin cells. The current opinion holds that transcription factor Ascl1 (Mash1) is the most likely initiator of catecholaminergic differentiation of chromaffin cell progenitors [20]. Ascl1 regulates the differentiation of neuroblasts from the neuroepithelium of the neural tube, as well as the subsequent formation of the autonomic nervous system [21]. Studies have shown that, in animals deficient in the Ascl1 gene, the adrenal medulla is formed, but chromaffin cells do not synthesize catecholamines and do not have a typical secretory phenotype [20]. Transcriptional regulation of chromaffin cell development will be described in more detail below.

A key stage in the differentiation of chromaffin cells is the initiation of the synthesis of tyrosine hydroxylase, an enzyme that catalyzes the first reaction of catecholamine synthesis—the conversion of tyrosine to L-dihydroxyphenylalanine. In the rat embryo, tyrosine hydroxylase expression is reliably detected on the 18.5th day of embryonic development and increases by the time of birth [22]. In mice, the first signs of tyrosine hydroxylase immunoreactivity are detected in the adrenal glands already on the 12.5th day [23]. On the 13.5th day, tyrosine hydroxylase-positive chromaffin cells colonize the inner adrenals and then aggregate, forming the medulla, on the 14.5th day. It is likely that settling the medulla is a long-term process and takes place not only in prenatal development, since single tyrosine hydroxylase-positive chromaffin cells are detected in the cortex of newborns. Chromaffin cells are subsequently divided into two types: A cells, which produce epinephrine, and NA cells, which synthesize norepinephrine. The former are distinguished by the expression of the phenyl ethanolamine-N-methyltransferase gene, an enzyme that catalyzes the conversion of norepinephrine to epinephrine. These cells make up 80–85% of the total number of chromaffin cells in the adrenal glands. According to available data, phenylethanolamine-N-methyltransferase is detected in the adrenal glands of rats on the 15.5–17th day of prenatal development [24]. Thus, at the end of the prenatal period of ontogenesis, the adrenal gland has a fully formed set of enzymes necessary for the synthesis of catecholamines.

## 3. Regulation of Chromaffin Cell Embryonic Development

### 3.1. Glucocorticoid Signaling

Adrenal cortical and chromaffin cells are closely related during embryonic and postnatal development. Adrenal glucocorticoids have been shown to promote the production of hormones by chromaffin cells, including regulating the expression of the enzyme phenylethanolamine-N-methyltransferase [25]. Higher expression of phenylethanolamine-N-methyltransferase in some chromaffin cells ensures the prevalence of epinephrine over norepinephrine production and makes the former the dominant catecholamine produced by the adrenal medulla in the postnatal period. In addition, glucocorticoids are necessary for the expression of the neuroendocrine secretory protein chromogranin A [26].

The in vitro studies of adrenal chromaffin and sympathetic ganglia progenitor cells have revealed that glucocorticoids suppress the development of the neuronal phenotype; in particular, they prevent the appearance of processes in sympathoadrenal progenitor cells [27]. However, further studies have shown that, in mice knocked out for the glucocorticoid receptor gene, adrenal medullary cells have a typical structure and express markers of chromaffin cells [26]. The absence of glucocorticoid signaling did not change the development program of cells but suppressed the expression of phenylethanolamine-N-methyltransferase, which made the cells noradrenergic [28]. The results of experiments on the ablation of the cortex show that the absence of both glucocorticoids and other factors synthesized by cortical cells does not significantly affect either the migration of chromaffin cell precursors or the development of a specific chromaffin phenotype [29,30]. Later investigations found that signaling of glucocorticoids is needed to maintain the population of chromaffin cells in the postnatal period of ontogeny. It is shown that the deletion of glucocorticoid hormone receptor genes leads to apoptosis of adrenal chromaffin cells and accelerates the death of chromaffin cells of extra-adrenal localization, and the introduction of glucocorticoid hormone drugs slows down the death of extra-adrenal chromaffin cells [31].

### 3.2. Transcriptional Regulation

In the embryonic period, a part of the neural crest cells migrates ventrally after delamination and reaches the dorsal aorta. The migration is ensured by ligand–receptor interactions mediated by bone morphogenetic proteins and the chemoattractant factor CXCL12, as well as class 3 semaphores and neuropilins [32]. Further, the expression of specific transcription factors, including Phox2b [33,34], Ascl1 [20,35], Insm1 [36], Hand2 [37,38], Gata2/3 [39,40], AP-2β (activating protein 2β) [41], and Islet-1, is activated in these cells [42]. With the exception of AP-2β, also known as TFAP2B, a transcription factor involved in various developmental processes, which is detected during the migration of neural crest cells, the expression of these genes is initiated shortly after the cells aggregate in the dorsal aorta in response to local signals of bone morphogenetic proteins-2,4,7 [29].

Phox2b (Paired-like homeobox 2b) is a homeodomain transcription factor associated with the development of neural crest derivatives and is highly specific for autonomous nervous system cells [43,44]. It has been identified as a major factor regulating the development of disease-related cells. In its absence, the development of both chromaffin cells and sympathetic neurons is completely blocked [33]. Other transcription factors determine cell development, including neuronal and endocrine differentiation, proliferation regulation, and apoptosis. Although the precursors of chromaffin cells and sympathetic neurons have a largely identical set of transcription factors during their early development, some of these factors have different levels of expression, and their functions may be different for these cell types.

The above-mentioned transcription factor Ascl1 (Achaete-scute homolog 1) plays an important role in various cellular populations of the central and peripheral nervous system during the embryonic period [34,45]. Ascl1 is characterized by time expression in sympathetic neurons and chromaffin cells, with the precursors of chromaffin cells expressing it for a longer period than differentiating sympathetic neurons. The absence of this factor leads to inhibition of the differentiation of Schwann cell progenitors into chromaffin cells [17]. Blockade of this transcription factor in mouse embryonic chromaffin cells leads to increased synthesis in them of several neural markers, including neurophyllum 68, and a lack of typical secretory granules, indicating a violation of neuroendocrine differentiation [20]. Activation of the expression of Ascl1 in differentiated chromophthalmic cells promotes their transdifferentiation into neuronal cells [46]. Studies of animals that do not express Ascl1 have shown that the absence of Ascl1 does not completely block the formation of chromaffin cells and sympathetic neurons, which allows us to assume that this transcription factor controls some transitional stage in the formation and regulates the rate of cell differentiation [20].

INSM1 (Insulinoma-associated protein 1) is a protein that plays a crucial role as a transcription factor in the differentiation of embryonic neuroendocrine cells. In situ hybridization studies of human tissues have shown the presence of the *INSM1* transcript in various regions of the brain, as well as in endocrine organs such as the pancreas, adrenal glands, thymus, thyroid gland, and endocrine cells of the gastrointestinal tract [47,48]. Mutation of the *INSM1* gene disrupts the terminal differentiation of chromaffin cells and the sympathoadrenal line as a whole. Consequently, it leads to a deficiency of catecholamines, which causes fetal mortality in mice with the Insm1 mutation [36]. At the same time, mice with INSM1 deficiency have a similar phenotype to Ascl-1-deficient animals, including delayed neuronal differentiation, reduced proliferation, and impaired differentiation of chromaffin cells, which suggests that Insm1 may be a downstream target gene of Ascl-1, while Ascl-1 activates endogenous Insm1 during chromaffin cell differentiation [49].

Hand2 (Heart- and neural crest derivative-expressed protein 2) is a transcription factor that ensures the embryonic development of the nervous system and the heart and, accordingly, the survival of the embryo as a whole. Mutations of the Hand2 gene do not disrupt the expression of the transcription factors Phox2b, Mash1, and Gata2/3; therefore, it is an independent regulator, contributing to the formation of the catecholaminergic phenotype in cells [38].

The family of zinc-finger transcription factors GATA accompanies the entire period of adrenal embryonic development. GATA4 expression begins at the earliest stages of development in the celomic epithelium and is considered a necessary condition for the development of the adrenogenital anlage and the initiation of the SF-1 factor [50,51]. GATA3 forms a noradrenergic secretory phenotype in cells. GATA3 mutations cause the absence of tyrosine hydroxylase expression, and consequently, the cells fail to synthesize catecholamines. This usually leads to the death of the organism in the embryonic period [52]. They also interfere with the expression of the Phox2b-dependent factor GATA2, the loss of which leads to early embryonic death [39]. In embryonic development, GATA2 appears after Ascl1, Phox2b, and Hand2 but before the synthesis of enzymes involved in catecholamine synthesis in cells.

Islet1 belongs to the family of LIM-homeodomain transcription factors that regulate the development of neuronal (sensory and motor neurons, retinal and heart neurons) and other cell populations [53,54,55]. The expression of Islet-1 in sympathoadrenal cells begins after they populate the dorsal aorta and is retained in the sympathetic ganglia and adrenal chromaffin cells, but at lower levels. In Islet1-deficient mice, the development of adrenal chromaffin cells suffers to a lesser extent than that of sympathetic neurons. Studies show that Islet1 deficiency blocks the expression of the phenoxyethanol-N-methyltransferase gene and has virtually no effect on the synthesis of tyrosine hydroxylase and dopamine-β-hydroxylase, i.e., it regulates the late stages of the development of chromaffin cells [42].

The family of transcription factors AP-2 regulates the processes of proliferation and differentiation of cells in embryonic development. AP-2β is a coordinated factor of Islet1, which is also necessary for the survival of the subpopulation of sympathetic neurons [41]. It is required for endocrine differentiation and maturation of chromaffin cells, including the proper formation of chromaffin vesicles and regulation of the expression levels of phenoxyethanol-N-methyltransferase [56].

The development of the adrenal medulla is also regulated by the Sox family of transcription factors. The Sox family comprises well-established regulators of cell fate decisions during development. The expression of the Sox10 protein from the SoxE group occurs in the migrating cells of the neural crest and is a prerequisite for the initiation of the sympathoadrenal cells development [57]. Sox4 and Sox11 were identified as important regulators of the proliferation and survival of sympathetic neuron progenitors. Their expression begins after the activation of the transcription factor Phox2b [58].

Thus, the transcriptional regulation of the adrenal chromaffin cell embryonic development is quite complex and involves multiple factors. Chromaffin cells share patterns of transcriptional regulation with neurons at early stages of development, but obviously have distinct regulation at later stages of differentiation. Less is known about proliferation control, especially in the postnatal life, which ensures regeneration and cell turnover.

## 4. Sources of Chromaffin Cell Postnatal Maintenance and Reparation

### 4.1. Proliferating Cells in the Adrenal Medulla

Despite the limited number of scientific reports on chromaffin cell proliferation, the ability of chromaffin cells in the adrenal to divide during postnatal ontogeny is well recognized. Evaluation of mitotic activity in the newborn mice has revealed that the total cell cycle of adrenal chromaffin cells lasts 7 h, divided into an S-phase of 1.5 h, a G2-phase of 1 h, an M-phase of 4 h, and a G1-phase of 0.5 h [59,60]. Proliferation slows down with age, but even adult animals show dividing chromaffin cells in the adrenal medulla [60,61].

Determination of mitotic activity parameters of chromaffin cells in postpubertal animals by evaluation of scientific data is complicated by numerous differences in strain, sex, age of animals, and, most importantly, different methods of counting cell proliferation. The first attempt to assess the proliferation rate of the adrenal chromaffin cells was made in 1919 by C.M. Jackson [62]. The investigation showed that chromaffin cells proliferate in young, but not in adult rats. Thirty years after R. Mitchell confirmed these results on the same strain of rats [60]. These data formed the basis for the concept that chromaffin cells in an adult organism are not capable of division and, consequently, of physiological regeneration. It is most likely that the absence of mitoses in the R. Mitchell’s and C. Jackson’s investigations was due to their visual search for cells in various stages of mitosis in histological slices under a light microscope. Other mitotic counts also revealed extremely low to absent numbers of dividing cells in the rat adrenal medulla [63,64]. However, later experiments of other researchers with the application of radio-immunometric techniques have revealed mitotic activity in the adult adrenal medulla [59,65,66,67]. In Table 1, we present a chronology of studies of the postnatal proliferation rate of the adrenal chromaffin cells in rodents.

The presence of proliferating cells has changed the idea of the postmitotic status of the adrenal chromaffin cells and put forward a hypothesis about the possibility of stimulating cell division. Proliferative response of the adrenal chromaffin cells was identified by Tischler A. et al., who demonstrated a dose-dependent increase in the number of dividing cells after administration of reserpine to 9-month-old Long–Evans and Sprague–Dawley rats [68,69]. The investigators also found that the proliferative response could be prevented by denervation of the adrenal gland, indicating that mitotic activity of the chromaffin cells is under sympathetic nervous system control [70].

**Table 1 ijms-26-09369-t001:** Detection of proliferative ability of the rodent adrenomedullary chromaffin cells in postnatal life.

Method Used	Rate of Cell Division	Animal, Strain	Age	Reference
Mitotic count	3 mitoses in 1 section No mitoses	Albino rat	56 days 340 days	Jackson C.M., 1919 [62]
Mitotic count following colchicine	Average of 3 mitoses per section No mitoses	Albino rats of both sexes	From birth to 63 days	Mitchell R. et al., 1948 [60]
[^3^H]thymidine	0.5% medullary cells labeled	Sherman albino rat	200 g	Messier B. et al., 1960 [65]
Mitotic count	Average of 20 dividing cells per adrenal (0.004%)	female Sprague–Dawley rats	22–36 weeks	Malvaldi G. et al., 1968 [63]
Light and electron microscopic autoradiography	9.4% in 2-week-old and less than 1% in adult	Mus musculus	From 2 weeks to adulthood	Jureska W. et al., 1978 [67]
Mitotic count following colcemid	0.3% of the cells divide	Long–Evans and Sprague–Dawley rats	9 months	Tischler A. et al., 1988 [68]
[^3^H]thymidine autoradiographia	Fluctuating mitotic rate from 1 to 70%	CS1 mice	Newborn within 24 h	Monkhouse W. et al., 1988 [59]
Mitotic count following Colcemid BrdU 6 h before Killing	1 cell or less per section Up to 7 labeled cells per section	Long–Evans and Sprague–Dawley rats	9 months	Tischler A. et al., 1989 [69]
Percentage of metaphases	0.5–0.8%	Wistar rats	300–350 gg (12–15 weeks)	Plecas B. et al., 1990 [64]
BrdU 50 μg/kg i.p. BrdU 1 μg/kg/h i.p.	0.92 + 0.09% after 192 h 41.96 + 1.43 after 1752 h	Sprague–Dawley rats	22–36 weeks	Verhofstad A., 1993 [66]
Ki-67 immunohistochemical evaluation	2.11 ± 0.15% in 6-week-old 1.65 ± 0.10 in 10-week-old	Wister rat	6–10 weeks old	Yaglova N.V. et al., 2018 [61]

Numerous scientific studies show that chromaffin cell proliferation in the adrenal medulla is a complex process regulated by a variety of factors including growth factors, neurogenic signals, hormones, and transcription factors (Figure 1). Proliferation of chromaffin cells has been found to demonstrate autocrine and paracrine regulation by growth factors, including fibroblast growth factor-2, nerve growth factor, and insulin-like growth factor-2 [28]. Besides self-produced cytokines, chromaffin cells seem to be controlled by adrenocortical cell cytokines. Adrenocortical epidermal growth factor (EGF) and Leukemia inhibitory factor (LIF) are considered putative modulators of chromaffin cell division. PC12 cells, an immortalized cell line established from a rat pheochromocytoma, demonstrate enhanced proliferation in response to EGF [71]. The presence of binding sites for EGF on normal chromaffin cells suggests the implication of EGF in their self-maintenance [72]. Leukemia inhibitory factor (LIF) is a pleiotropic cytokine that belongs to the IL-6 family. LIF is known to ensure the development and maintenance of the hypothalamic–pituitary–adrenal axis and chromaffin cells [73,74,75]. Adrenocortical cells also produce factors that inhibit the proliferation of the chromaffin cells. Zona reticularis, the inner layer of the adrenal cortex, produces the androgen hormone dehydroepiandrosterone, which has been shown to suppress adrenomedullary cell proliferation in vitro and in vivo [76,77,78]. Transforming growth factor-beta (TGF-β), secreted by developing adrenal chromaffin cells, also demonstrates antiproliferative action [76].

Less is known about transcriptional regulation of chromaffin cell division throughout postnatal life. Some investigations have shown that in adults, loss of the GATA3 transcription factor, essential for embryonic development of the chromaffin tissue and crucial for survival of postnatal neurons, does not affect survival of the adrenal chromaffin cells [21,39]. In our previous studies, we found expressions of the transcription factor PRH (the proline-rich homeodomain protein) in the adrenal chromaffin cells of adult rats [79]. PRH, also known as hematopoietically expressed homeobox (HHEX), is a DNA-binding protein that regulates embryonic development and cell proliferation and survival in adults [80,81,82]. Evaluation of PRH expressions in chromaffin cells in vivo revealed a negative association with their proliferation rate as well as in adrenocortical cells [83,84]. The findings suggest that the adrenal cortex and medulla share mechanisms of cell proliferation control in postnatal life. Despite the evident ability to divide in response to endogenous and exogenous influences, proliferation of the adrenal chromaffin cells is traditionally considered a weak source of regeneration.

### 4.2. Stem/Progenitor Cells in the Adrenal Medulla

The presence of stem cells in adult organs as a source for maintenance of tissue homeostasis is generally recognized. Reparation and renewal processes associated with adult stem/progenitor cells have been shown in various tissues including nervous ones [32,85]. The adrenal glands were also shown to maintain a population of progenitor cells in the subcapsular compartment [86,87]. Differentiation and centripetal migration of these progenitors ensure renewal of cortical steroid-producing cells [88,89]. Unlike the adrenal cortex, the presence of stem/progenitor cells in the medulla is still an open question. Earlier in vitro studies identified the cells with progenitor characteristics in bovine and human medullary extracts [90,91]; these cells could also generate spheres expressing progenitor cell markers such as CD133 (marker of primitive hematopoietic stem and progenitor cells), Nestin (a type IV intermediate filament protein expressed in neural stem cells), and Notch1, stimulating astroglial differentiation of neural progenitor cells. The Nestin-positive cells were negative for markers of differentiated chromaffin like tyrosine hydroxylase and chromogranin A.

The source and localization of progenitor cells capable of differentiation into chromaffin cells is also poorly understood. Some investigations showed reparation of the adrenal medulla after damage which was considered due to differentiation of pre-existing, primitive, undifferentiated, chromaffin cells, rather than mitosis [92,93]. Identification of Schwann cell progenitor origin of chromaffin cells makes them a putative source for self-maintenance of chromaffin tissue [94]. But the presence of multipotent Schwann cell progenitors retaining the capacity to regenerate autonomic neurons and chromaffin cells, in adult tissues, is still hypothetical and requires further investigation.

## 5. New Findings in Chromaffin Population Renewal

The adrenal medulla is composed of chromaffin cells, ganglion cells, and sustentacular cells. All of them represent derivatives of the neural crest [95]. Sustentacular cells are found in the adrenal medulla both in embryonic and postnatal life [96]. They have cytoplasmic processes that form a network within the adrenal medulla. Immunohistochemical investigations found expression of S-100 protein, a marker of glial cells [97]. Electron microscopy studies found that the fine structure of sustentacular cells closely resembles glial cells of the peripheral nervous system [98]. Unlike the first two, the sustentacular cells represent a small population with a still debatable function. Their proximity to chromaffin cells has led to their name, since the formation of a network by the sustentacular cells indicated a most likely supportive function. Sustentacular cells are considered non-endocrine. Identification of ion traffic between sustentacular and chromaffin cells confirmed their supporting function [99]. Moreover, sustentacular cells have been found to regulate calcium metabolism, and indirectly, the release of catecholamines since exocytosis of secretory granules is triggered by increases in intracellular calcium concentration [99]. Further investigation revealed participation of nestin-positive glia-like progenitor cells in stress response and contribution to the generation of chromaffin cells and neurons [100]. Genetic tracing and transcriptomic data of the mouse adrenal medulla have found a subpopulation of cells expressing pluripotency factor SOX2 in the postnatal mouse adrenal medulla, able to give rise to chromaffin cells [101].

Another source of chromaffin cell renewal has been found in investigations of changes in transcription control during the transition of the rat adrenal gland from growth to a period of self-maintenance. The rat adrenals increase in size until the 11th week of postnatal life [102]. Then, the growth stops, the proliferative activity of its cells decreases, and the mechanisms of cell renewal should be activated.

We conducted an immunohistochemical study to search for early progenitor cells expressing the transcription factor Ascl1, which ensures the differentiation of Schwann cell progenitors into pheochromocytes. Cells synthesizing this factor in detectable quantities in sections of the adrenal glands were not found either during their growth period or after its completion. Moreover, all pheochromocytes actively expressed tyrosine hydroxylase, a marker of a highly differentiated chromaffin cell [103].

Next, we investigated the dynamics of activation of canonical Wnt signaling. Wnt-signaling affects many aspects of the development and function of the nervous system [104]. Wnt proteins activate a variety of signaling pathways and induce a variety of processes, including cell proliferation and differentiation, changes in gene transcription, cytoskeleton rearrangement, and others. We have demonstrated that β-catenin is expressed by chromaffin cells of the adrenal glands in postnatal development. It should be noted that with age, the total number of β-catenin-positive cells does not change, but the number of cells with translocation of β-catenin into the nucleus after the completion of organ growth increases; that is, after the growth stops, the activation of canonical Wnt signaling in chromaffin cells increases [103]. These findings suggest induction of cell division and differentiation.

The Sonic hedgehog (Shh) signaling pathway is known to play an important role in neurogenesis and the formation of neural patterns during the development of the nervous system [105,106]. It was previously found that mechanically constitutive activation of Shh signaling correlates with the activity of Wnt signaling [107]. Shh expression is activated in neurons during ischemia/hypoxia, and inhibition of the Shh pathway exacerbates ischemic damage to brain neurons in rats, indicating the role of Shh in the regeneration of nervous tissue [108,109]. In our developmental research, we found a low number of chromaffin cells with Shh nuclear localization, and a significant increase in Shh-positive cells by the time of completion of adrenal growth [103].

An important issue of cell renewal investigations is the postnatal expression of pluripotency factors. Less is known about the ability of chromaffin cells to express embryonic pluripotency factors during ontogeny. Transcription factors POU5F1 or Oct4 and Sox2 are expressed at the earliest stages of mammalian embryogenesis and control maintenance of the pluripotent state [110]. Sox2 is also crucial for maintaining pluripotency, but is not required for its establishment during embryogenesis, possibly due to duplication with other members of the Sox family [111]. In our experiments, we revealed synthesis and translocation to the nucleus of the transcription factor Oct4 in the chromaffin cells of the adrenal medulla during postnatal development. Moreover, the percentage of Oct4-positive cells in chromaffin cells increased after completing growth [112]. The completion of the adrenal growth period was accompanied by a natural decrease in the number of proliferating cells, whereas expression of tyrosine hydroxylase remained at a high level and was observed in pheochromocytes, indicating their terminal differentiation and significant functional activity during the growth period and after it declines. This means that the identified Ki-67-, β-catenin-, Shh-, and Oct4-positive cells are highly differentiated chromaffin cells. The presence of Oct4- and Shh-positive cells, therefore, suggests their possible dedifferentiation. Verification of the hypothesis that upregulation of expression of the above-mentioned factors is associated with transition from growth to the self-sustaining phase of the chromaffin cell population requires the application of a morphogenesis dysregulation model.

We used an original model based on our studies of endocrine disruptors, substances capable of interfering with any stage of the synthesis of hormones and their interaction with target cells, acting at low doses similar to physiological concentrations of endogenous hormones [113]. Dichlorodiphenyltrichloroethane (DDT) was used as an endocrine disruptor. Exposure to its low doses begins from conception and lasts throughout prenatal and postnatal development [114]. Before the puberty period, the adrenal medulla develops at a normal rate, but with the onset of puberty, the proliferative activity of chromaffin cells decreases by the postpubertal period, resulting in smaller brain matter. And by the 70th control day, when the growth of the rat adrenal glands is to be completed, conversely, the activation of the growth processes begins due to the increase in proliferative activity of the cells [115]. That is, the model was slowing down and accelerating morphogenetic processes.

The study showed that during the period of adrenal growth inhibition, there is a tendency to increase the activation of canonical Wnt-signaling and a significant increase in the number of Sonic Hedgehog-positive and Oct4-positive chromaffin cells [84]. It is noteworthy that all these cells also expressed tyrosine hydroxylase. However, early chromaffin cell progenitors expressing the transcription factor Ascl1, specific to Schwann cell precursors, were not found. Sox2-positive cells were not observed either [116].

The growth activation was accompanied by a decrease in the number of cells synthesizing the anti-proliferative factor PRH and a decrease in the number of cells with activation of canonical Wnt signaling compared to the age control. In the growing medulla, a significantly smaller number of Oct4- and Sonic Hedgehog-positive cells was detected. Low-differentiated precursors were also absent, and all chromaffin cells actively expressed tyrosine hydroxylase [84,116]. The obtained data indicate that the end of growth processes is accompanied by the creation of a pool of cells among highly differentiated chromaffin cells capable of changing their fate.

Interestingly, adult chromaffin cells also demonstrate phenotypic plasticity. Intrabrain transplantation of chromaffin non-dopaminergic cell aggregates of Zuckerkandl’s organ was shown to enhance the dopamine levels in grafted striatum and improve functional deficits in Parkinsonian rats, indicating transformation of adrenergic to dopaminergic cells [117]. Changes in secretory phenotype may possibly result from the existence of sympathoadrenal progenitors with the potential to differentiate to the endocrine and neural lineages within the adult adrenal medulla [118]. Improving the efficiency of converting chromaffin cells to neuronal phenotypes and ensuring their functional integration into the host brain is an ongoing area of research.

## 6. Discussion

Cells of the nervous tissue are known to vary in repair and renewal rates. Neurons fail to renew the population, whereas glial cells demonstrate a higher ability to repair and regenerate [119]. The earlier concept of chromaffin cells originated from neuronal progenitors, and the postmitotic state of these cells in postnatal life was logical but inconsistent with growing evidence of their proliferation. The experimental data show that adrenal chromaffin cells are capable of both proliferation and proliferative responses to various triggers, and, therefore, of regeneration and self-renewal. Ability to repair and renew advocates for the current opinion that chromaffin cells originate from Schwann cell progenitor cells and do not represent modified neurons from an embryological point of view. Differences in transcriptional regulation also suggest another pathway of differentiation. The absence of changes in chromaffin cells’ survival, unlike neurons in adult mice with the deleted transcription factor Gata3, also indicates different embryonic progenitors of these cell populations [39]. Lesser disorders in embryonic development of adrenal chromaffin cells compared to sympathetic neurons in Islet1-deficient mice also show divergent embryogenesis of these two cell populations [42]. Nowadays, there is no compelling evidence that transcription factors controlling the proliferation and differentiation of chromaffin cells in the embryonic period perform their function during ontogeny. The network of regulating factors and cellular and molecular mechanisms of self-maintenance are yet to be detected. But the recent findings clearly show that the adrenal chromaffin cells do not persist as a stable population and also undergo transition of developmental stages in postnatal life. The ability to activate morphogenetic processes allows us to assume that the chromaffin cells have a regenerative resource.

Analysis of currently available scientific data allows us to formulate three basic most probable mechanisms of physiological chromaffin cell regeneration: by preserving some spontaneously dividing cells, by differentiating the multipotent precursors of Schwann cells/sustentacular cells, and by creating a pool of highly differentiated pheochromocytes, ready for further dedifferentiation (Figure 2). It is possible that spontaneously proliferating chromaffin cells also represent dedifferentiating cells. These mechanisms, especially dedifferentiation of cells, may also participate in the altered functional profile and acquisition by chromaffin cells of a dopaminergic phenotype.

## 7. Conclusions and Future Directions

The presence in adult medulla pools of cells with activated expression of pluripotent factors, morphogenic pathways, and low-differentiated progenitors implies multiple regeneration mechanisms and requires further research to establish their participation and activation sequence in the processes of reparation and self-maintenance in the postnatal period. Considering the triple way of cell renewal allows the development of various approaches to restore chromaffin cells and prolong their survival. Transformation of chromaffin cells into neuronal phenotypes and ensuring their survival is also a promising area of research. Investigations in this area will obviously make a significant contribution to the cellular therapy of neurodegenerative diseases. The identified events occurring in the postnatal life of the chromaffin cells, ways to restore and maintain their population, and interaction of various cell types in the adrenal medulla, show that even cells with low regenerative potential may have overt and covert mechanisms, the activation of which can restore cell populations. Elaboration of approaches to induce and manage these mechanisms will allow for maintaining cultured cells in vitro, and obviously make a significant contribution to practical regenerative medicine.

## Figures and Tables

**Figure 1 ijms-26-09369-f001:**
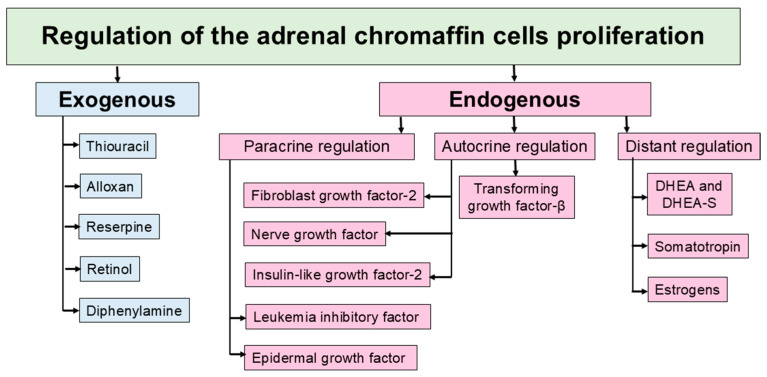
Factors regulating the adrenal chromaffin cell proliferation.

**Figure 2 ijms-26-09369-f002:**
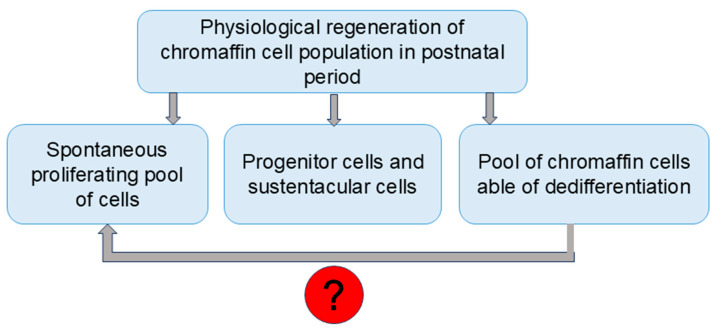
New concept of chromaffin cell physiological regeneration in postnatal period. “?” indicates possible association of dedifferentiation with further proliferation.

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
