# Peer review of "New Concepts of Regeneration and Renewal of Adrenal Chromaffin Cells"

_ijms, 2025, doi:10.3390/ijms26199369_

Round 1

Reviewer 1 Report

Comments and Suggestions for Authors

I suggest to add more specific investigations (already done or predicted) of chromaffin regeneration or grafting, and how they could change the future of people with neurodegenerative diseases, or even prevent them.

I also recommend to discuss if regenerated chromaffin cells could suffer a modification in their stimulus-secretion coupling, and their calcium metabolism.

Comments on the Quality of English Language

The use of written English is, in some parts, confusing. I think they lack some prepositions and, in some phrases, they write incomplete or not clear ideas.

For example this phrase: "Identification of ion traffic between sustentacular and chromaffin cells"

Reviewer 2 Report

Comments and Suggestions for Authors

This manuscript by N. V. Yaglova et al., is a narrative review on a highly conflicting issue i.e., whether the adult chormaffin cells of the adrenal medullary gland have the potential for regeneration.

Along the last century, an overwhelming literature has supported the view that sympathetic neurons and adrenal medullary chromaffin cells have a common origin, the neural crest. Additionally, from a physiological point of view both cell types tightly collaborate in the triggering of the fight-or-flight response during stressful conflicts, a mammals sudden reaction mediated by catecholamine, first defined by W. Cannon in 1932. All elements of the catecholamine synthetic, storage, and release machineries are common to both sympathetic neurons and chromaffin cells, that form the unique sympatho-adrenal axis.

In their Introduction, authors mention the various attempts made in the past decades to use chromaffin cells as donors in animal models and humans with Parkinson’s disease, stroke, brain injuries, or chronic pain. This was based in the conjecture that grafted chromaffin cells will act as donors of neurotransmitters, growth factor, or opiates. These therapeutic attempts have been a failure.

Author then extensively describe the origin and regulation of chromaffin cell embryonic development to end with the evidence supporting the proliferative ability of the rodent adrenomedullary chormaffin cells, which is summarized in the extensive table 1. Of note is the fact that this table contains quite old studies and only one study from 2018. This is an indicaction of the poor interest that this theme arises actually.

I believe that Author have made a substantial effort to support their hypothesis on chromaffin cell regeneration, but only with scarce data and mostly from their laboratory. May be they should be more cacious in their claims.

Reviewer 3 Report

Comments and Suggestions for Authors

The review submitted by Yaglova and co-authors entitled “New concepts of regeneration and renewal of adrenal chromaffin cells” broaches an interesting topic that has evolved a great deal in the last thirty years or so. Nearly every sentence needs to be rewritten in view of making it more direct and precise.

The review is organized with subheadings into an Introduction (see below for critique), a section on cell origins, a section on embryonic maturation divided into response to glucocorticoids and another to various transcription factors

Specific points to address:

  1. Part 1, Introduction, does not introduce as much as state obvious generalities, without providing details. Why are we interested in self-renewal of chromaffin cells? To what therapeutic uses could they be put? If they are only the main source of catecholamines, what are the minor sources, and are they systemic or local? What is the functional similarity to adrenergic neurons? If a large number of transplantations have been carried out over the last fifty years, why are only two cited? Chromaffin cells are considered to be “low-proliferating” [sic] by whom? What does it matter? Who had the point of view that they are post-mitotic with what evidence? Which later investigations refuted the concept and with what evidence? And so on. A review proposes to summarize such work and point the reader in the direction of the studies that are evaluated. The citation to “the scientific literature” about postnatal proliferative ability dates from 1993. Finally, at the end of an entire page of text, the reader can finally rest assured that NOW the authors will broach “the available data on adrenal chromaffin cell origin and embryonic development, postnatal proliferation and self-maintenance and novel concepts of chromaffin cell reparation.” Except that it’s not clear if the reader can trust the authors to have really looked at all available data, since there is very little data
  2. The authors are encouraged to remove Figure 2, which is useless, and replace it with a schematic of the various paracrine influences on, and transcription factor-mediated responses within, chromaffin cells over time. Islet1 is not “a family of… factors” but rather, one member of the small family. Isl1 is expressed and critical for many other cell types. How did the authors evaluating deficient mice assess chromaffin development in the light of these other defects? The presentation of such important factors as Sox or GATA family members was too superficial and like reading a shopping list.
  3. Line 77, the authors mention work on “prospongioblasts” but never talk about it again. What evidence is there that Dr Smitten was correct, for them to even broach the topic? In essence, it is helpful in a historic review as this had set out to be, to review the evidence available at a given time. 
  4. The problems in the language are linked to reasoning. For example, at line 99, the authors state “studies have shown” and then not highlighted an actual study. Which original study showed that claim and on what credible evidence? Why did they broach this after a sentence beginning “The current opinion [of whom?] holds that…” and then cite a review from 2002 that is supposed to summarize the groupthink of the worldwide researchers who may have opinions about this?
  5. Line 368, it would be interesting to hear more about the significance of the ion traffic between sustentacular and chromaffin cells. What is the purpose of such traffic, in the view of the authors, and again, with what evidence that convinced them of this opinion?
  6. The authors should tell the readers more about their critical views on the most current work (and not only their own), such as reference 103. They only write about it that “Genetic tracing and transcriptomic data of the mouse adrenal medulla have found a subpopulation of cells expressing pluripotency factor SOX2 in postnatal mice adrenal medulla, able of giving rise to chromaffin cells [103].” So why do adrenal medullas mostly not self-renew in vivo, in their informed perspective?
  7. The authors should begin a new paragraph at line 423 when they discuss their own work in the context of this review. If I understand it properly, they expose postnatal rats to "environmental" DDT doses for at least 70 days, and instead of seeing a typical mature adrenal gland, they still find proliferation. They should explain what data led them to conclude this is "slowing down and accelerating morphogenetic processes" or remove this sentence (the preferred solution).
  8. They should cite the work at line 441 that asserts that Ascl1 is a transcription factor specific for Schwann cell precursors. And at line 447, what are "low-differentiated" precursors and how were they sought in the tissue (with what markers?) What was the expected control outcome in looking for them?
  9. As a developmental biologist, I disagree with the assertion that [sic] "Ability to activate morphogenetic processes allows to assume that the chromaffin cells have regenerative resource."
  10. This review paper needs a great deal of work to turn it into a reliable and understandable summary of the field that the authors set out to discuss. I suggest they also include some some non-text-based illustrations.
Comments on the Quality of English Language

Unfortunately, some of the nuanced meaning in the overview may have been lost in translation, which occasionally hinders clear communication on this complex subject
